# Von Willebrand Factor Collagen-Binding Activity and Von Willebrand Factor-Mediated Platelet Adhesion in Patients with Coronary Artery Disease

**DOI:** 10.3390/biomedicines12092007

**Published:** 2024-09-03

**Authors:** Zufar Gabbasov, Sergey Okhota, Yuliya Avtaeva, Olga Saburova, Ivan Melnikov, Valentina Shtelmakh, Sergey Bazanovich, Konstantin Guria, Sergey Kozlov

**Affiliations:** 1Laboratory of Cell Hemostasis, Chazov National Medical Research Centre of Cardiology of the Ministry of Health of the Russian Federation, 121552 Moscow, Russia; julia_94fs@mail.ru (Y.A.); saburovaos@mail.ru (O.S.); ivsgm@mail.ru (I.M.); bazarus13.ukr@yandex.ru (S.B.); kgguria@gmail.com (K.G.); 2Department of Problems of Atherosclerosis, Chazov National Medical Research Centre of Cardiology of the Ministry of Health of the Russian Federation, 121552 Moscow, Russia; ae2007@mail.ru (S.O.); valyashtelmakh13@mail.ru (V.S.); bestofall@inbox.ru (S.K.); 3Laboratory of Gas Exchange, Biomechanics and Barophysiology, State Scientific Center of the Russian Federation, The Institute of Biomedical Problems of the Russian Academy of Sciences, 123007 Moscow, Russia

**Keywords:** von Willebrand factor, collagen, glycoprotein Ib, platelet adhesion, coronary artery disease

## Abstract

In this study, we investigated von Willebrand factor (VWF)-related parameters in 30 patients with stable coronary artery disease (CAD) and 50 patients without CAD. In both groups, the following were measured: the VWF antigen level (VWF:Ag); the VWF ristocetin cofactor activity (VWF:RCo); the VWF collagen-binding activity (VWF:CB); and VWF-mediated platelet adhesion. Platelet adhesion was measured in whole blood at a shear rate of 1300 s^−1^ using a microfluidic chamber with a collagen-coated surface. VWF:Ag and VWF:RCo were found to be the same in both groups of patients. However, VWF:CB was found to be lower in patients with CAD compared with patients without CAD, with values of 106.7% (82.1; 131.6) and 160.4% (112.5; 218.1), respectively (*p* < 0.001). The decrease in platelet adhesion after GPIb inhibition was more pronounced in patients with CAD compared with patients of the control group, with recorded values of 76.0% (60.6; 82.1) and 29.3% (0.0; 60.4), respectively (*p* < 0.001). After adjusting for traditional risk factors, the odds ratio for CAD was found to be 0.98 (95% CI, 0.97–0.99; *p* = 0.011) per 1% increase in VWF:CB activity, and 1.06 (95% CI, 1.03–1.09; *p* < 0.001) per 1% decrease in GPIb-mediated platelet adhesion. The findings presented in this paper indicate a possible critical role played by complex VWF–collagen-platelet interactions in the development of CAD.

## 1. Introduction

Coronary artery disease (CAD) is among the leading causes of disability and mortality worldwide, accounting for 17.8 million deaths annually [1].

Von Willebrand factor (VWF) is a large multimeric glycoprotein that plays an important role in hemostasis. VWF synthesis in vivo occurs in endothelial cells or megakaryocytes. Depending on the site of synthesis, VWF accumulates either in endothelial cells or in platelet α-granules. After damage to the vascular wall, soluble agonists trigger the release of VWF from the endothelium and α-granules of platelets; VWF then interacts with platelet glycoprotein Ib (GPIb) to stimulate platelet adhesion and aggregation [2]. VWF deficiency or dysfunction often leads to bleeding. Contrarily, an increase in the concentration of high-molecular-weight VWF multimers is closely associated with the development of arterial thrombotic complications [3]. Elevated levels of VWF antigen (VWF:Ag) in blood plasma have often been associated with cardiovascular diseases [4]. However, a number of authors have assumed that measuring the VWF:Ag level in addition to traditional risk factors does little to aid the prediction of acute cardiovascular events [5].

VWF multimers are sensitive to shear rates [3]. They are activated at shear rates that can occur at sites of arterial stenosis. Activated VWF multimers bind to the collagen of the damaged arterial wall and facilitate platelet adhesion through the binding of VWF to the GPIb-IX-V complex of platelet receptors. Thus, VWF can serve as an active bridge between platelets and the subendothelial matrix. We may then hypothesize that the functional disturbance of VWF and the changes in the binding activity of this protein with platelets affects thrombus formation at the initial stage and thus contributes to the development of CAD.

In this study, we assessed VWF:Ag levels, VWF activity using VWF-ristocetin cofactor activity (VWF:RCo), VWF collagen-binding activity (VWF:CB), and platelet adhesion to collagen under flow conditions in patients with and without CAD. To our knowledge, VWF-mediated platelet adhesion and VWF:CB were not investigated previously in CAD patients.

## 2. Materials and Methods

### 2.1. Study Design

This study comprised 30 patients with stable CAD. These patients had a history of or typical symptoms of CAD and underwent coronary angiography (CAG), which revealed stenotic lesions in the coronary arteries. The control group included 50 patients without clinical manifestations of CAD, who underwent CAG due to other diagnostic reasons. They had no stenotic coronary atherosclerosis detected via CAG and/or computed tomographic angiography of the coronary arteries. A lesion was considered stenotic if it resulted in a reduction of 50% or greater in the lumen of the left main, left anterior descending, left circumflex, and right coronary arteries, or a second-order branch with a diameter > 2 mm [6].

This study did not include patients with any of the following: unstable angina; within the first 2 months after myocardial infarction (MI), bypass grafting, or coronary angioplasty; familial hypercholesterolemia; low-density lipoprotein (LDL) cholesterol > 4.9 mmol/L; NYHA class III–IV chronic heart failure; left ventricular ejection fraction less than 40%; permanent atrial fibrillation/flutter; aortic or mitral stenosis; hereditary or acquired coagulopathies; malignant neoplasms; or clinical and laboratory signs of acute infectious disease within the previous two months.

The presence of traditional risk factors for CAD was assessed in all patients, and the results obtained for the two study groups were compared. The study design was the same as in our previous work [7].

### 2.2. Ethical Approval

This study followed the standards of good clinical practice and was carried out in line with the principles set out in the Declaration of Helsinki. In addition, this study was approved by the Ethics Committee of the Chazov National Medical Research Centre of Cardiology of the Ministry of Health of the Russian Federation, Moscow, Russia. Written informed consent was obtained from all participants.

### 2.3. Blood Sample Collection

Blood samples were collected from the cubital veins of participants. The blood sample collection and storage protocol was the same as in our previous work [7].

### 2.4. Determination of VWF:Ag Levels, VWF:RCo Activity, and VWF:CB Activity

The levels of VWF:Ag in patients were measured using an enzyme-linked immunosorbent assay (ELISA) (Thermo Fischer, Waltham, MA, USA). The VWF:Ag levels were expressed as a percentage of the normal content (ranging from 50 to 150% of the normal value).

VWF:RCo activity was determined by measuring the agglutination of formaldehyde-fixed human platelets using the classic aggregometer method. The method is based on the ability of VWF to induce platelet agglutination in the presence of the antibiotic ristocetin, which induces the binding of platelet GPIb receptors to the A1 domain of VWF [8,9]. The formaldehyde-fixed platelets, ristocetin, plasma calibrators, and control plasma were obtained from RPA RENAM (RPA “RENAM”, Moscow, Russia). The values obtained were expressed as a percentage of the normal content (ranging from 50 to 150% of the normal value).

VWF:CB activity was measured using a TECHNOZYM enzyme-linked immunosorbent assay (Technoclone GmbH, Vienna, Austria). VWF:CB values were expressed as a percentage of normal content (ranging from 50 to 250% of the normal value).

### 2.5. Measurement of Platelet Adhesion to the Collagen Surface

A microfluidic device for recording the kinetics of blood cell adhesion to a protein-coated surface under controlled flow conditions was described in a previous paper [10]. Briefly, this device consists of a flow chamber with an optically transparent collagen-coated surface, a peristaltic pump ensuring blood movement through the flow chamber, a laser, a photodetector, and an analog-to-digital converter connected to a computer [7]. This device was used in the present study. The first measurement was performed to obtain a baseline value of platelet adhesion. Then, to assess the influence of GPIb inhibition on platelet adhesion, a separate experiment was conducted on the blood of each patient; this involved the addition of 10 µg/mL of monoclonal antibody (mAb) AK2 against GPIb receptors, with a further 15 min period of blood circulation through a new flow chamber. To assess the influence of platelet activation inhibition on platelet adhesion, a separate experiment was again conducted on the blood of each patient; this involved the addition of 1 µmol/L of prostaglandin E1 (PGE1) with a further 15 min period of blood circulation through a new flow chamber. The results of the measurements in the two groups of patients were then compared. Typical curves of time-dependent increases in the intensity of scattered laser light recorded during 15 min periods of circulation of whole blood samples are presented in Figure 1.

The surfaces of the microfluidic device were coated with type I rat collagen (Sigma-Aldrich, St. Louis, MO, USA) via incubation with a 0.1 mg/mL collagen solution in phosphate-buffered saline (Sigma-Aldrich, St. Louis, MO, USA) for 2 h at room temperature. Prior to coating, the slides were rinsed with 70% ethanol and then dried. The monoclonal antibody AK2 against the ligand-binding domain GPIb of the platelet membrane GPIb-IX complex was a gift from Dr. A. Mazurov, of the National Medical Research Center of Cardiology. The collagen solution was stored at 4 °C, and the mAb was stored at −70 °C. The protocol for the collagen coating of optical surfaces was the same as in our previous work [7].

### 2.6. Statistical Analysis

The quantitative data collected in the study were presented as medians and quartiles (lower quartile, upper quartile). The Shapiro–Wilk W-test was used to test statistical hypotheses regarding the distribution type. Different methods of nonparametric statistics were used for comparative analyses of the data from the two patient groups: Fisher’s exact test and the χ^2^ test with Yates’s correction were used for comparing qualitative characteristics; the Mann–Whitney U-test was used for comparing quantitative characteristics in two independent groups; and the Wilcoxon test was used for comparing quantitative characteristics in two dependent groups. The associations between VWF:Ag, VWF:RCo, VWF:CB, platelet adhesion parameters, and CAD were assessed using logistic regression analysis and expressed as odds ratios (95% confidence interval (CI)). The *p*-value was significant at 0.05. The binary classification was evaluated using ROC analysis. The cut-off point for binarization was determined using the Youden’s J statistic. All tests were two-tailed. Statistical analysis was performed using Statistica v. 7.0 (StatSoft Inc., Tulsa, OK, USA) and SPSS Statistics v. 27.0 (SPSS Inc., Chicago, IL, USA) software.

## 3. Results

Patients with CAD were more likely to be male; they were also more likely to have diabetes mellitus, a low level of HDL cholesterol, or LDL cholesterol >3 mmol/L. Smokers were also more likely to have CAD (Table 1). MI as the first manifestation of CAD occurred in 53% of cases. Among patients with CAD, 79.7% had a history of coronary artery stenting and 13% had a history of coronary artery bypass grafting. Stenotic lesions of the left main coronary artery, the left anterior descending artery, the left circumflex artery, and the right coronary artery were detected in 11.2%, 79.7%, 56.2%, and 73.7% of cases, respectively.

In patients with CAD, the median 15 min platelet adhesion value was 7.9 mV (5.0; 11.7) at baseline and 1.8 mV (1.5; 2.7) after GPIb inhibition, representing a 76.0% (60.6; 82.1) relative decrease (*p* < 0.001). In control patients, the median 15 min platelet adhesion value was 12.6 mV (9.6; 15.7) at baseline and 10.3 mV (4.3; 14.9) after GPIb inhibition, representing a relative decrease of 29.3% (0.0; 60.4) (*p* = 0.001). The inhibition of GPIb leads to a greater reduction in platelet adhesion among patients with CAD, compared with the control group (*p* < 0.001) (Table 2). This study had sufficient power (0.99) to evaluate the difference in decreases in platelet adhesion after GPIb inhibition between the groups of patients.

Among patients with CAD, seven received only aspirin, five received only P2Y12 receptor inhibitors, and eighteen received dual antiplatelet therapy (DAPT). The absolute values of platelet adhesion were less pronounced in patients who received P2Y12 inhibitors or DAPT, compared with patients who received only aspirin. Nonetheless, antiplatelet therapy did not affect the relative decrease in platelet adhesion after GPIb inhibition. In the control group, 18% of patients received aspirin monotherapy, while others were free from any antiplatelet therapy. Aspirin monotherapy had no effect on the absolute values of platelet adhesion or the values of GPIb-mediated platelet adhesion. The same results were obtained in our previous study [9].

Among patients with CAD, the median 15 min platelet adhesion value after PGE1 addition was 6.8 mV (4.7; 10.3), representing a 3.9% (−44.0; 51.1) relative decrease (*p* = 0.50). Among control patients, the median 15 min platelet adhesion value was 12.6 mV (9.6; 15.7) at baseline and 5.9 mV (3.7; 7.9) after PGE1 addition, representing a 55.8% (40.7; 67.3) relative decrease (*p* = 0.001). The inhibition of platelet activation resulted in a greater reduction in platelet adhesion in patients from the control group compared with CAD patients (*p* < 0.001) (Table 3).

VWF:Ag levels and VWF:RCo activity were same in both groups. VWF:CB activity in the blood of patients with CAD was lower than in patients without CAD: 106.7% (82.1; 131.6) vs. 160.4% (112.5; 218.1), respectively (*p* < 0.001) (Table 4). This study had sufficient power (0.92) to evaluate the difference in VWF:CB activity between the groups of patients.

### 3.1. Univariate Logistic Regression Analysis

Univariate logistic regression analysis revealed that the following were all associated with CAD: a decrease in platelet adhesion after GPIb inhibition; VWF:CB activity; smoking; HDL cholesterol values of <1.0 mmol/L for males and <1.2 mmol/L for females; diabetes mellitus; and male sex (Table 5). Platelet adhesion at baseline was excluded from the analysis because it was confounded by patients’ antiplatelet medications.

The unadjusted odds ratio for CAD was 1.05 (95% CI, 1.03–1.07; *p* < 0.001) per 1% decrease in platelet adhesion after GPIb inhibition, and 0.98 (95% CI, 0.97–0.99; *p* = 0.002) per 1% increase in VWF:CB activity.

### 3.2. Multivariate Logistic Regression Analysis

The odds ratios obtained in the univariate analysis were adjusted for other risk factors and biomarkers using multivariate logistic regression analysis. The model included the following: a relative decrease in platelet adhesion after GPIb inhibition; VWF:CB activity; smoking; and age. Other risk factors and biomarkers were excluded because they did not affect or decrease the predictive value of the model. The model was significant (*p* < 0.001), with 86.3% of overall correct predictions for CAD. The adjusted odds ratio for CAD was 1.06 (95% CI, 1.03–1.09; *p* < 0.001) per 1% decrease in platelet adhesion after GPIb inhibition, and 0.98 (95% CI, 0.97–0.99; *p* = 0.011) per 1% increase in VWF:CB activity (Table 6).

Calculation was carried out using the following regression equation:$$F\left(z\right)=\frac{1}{1+e^{-z}}$$

Using the above equation, we obtained the following:Z = 0.084 × Age + 1.591 × Smoking − 0.019 × VWF:CB + 0.054 × Decrease in platelet adhesion − 6.184.

### 3.3. Receiver Operating Characteristic Analysis

A receiver operating characteristic (ROC) analysis was performed to assess the association of the logistic regression model and independent variables with CAD. The area under the curve (AUC) for the logistic regression model was 91.6% ± 3.1% (95% CI, 85.6–97.6%, *p* < 0.001). The best combination of sensitivity (90.0%) and specificity (84.0%) was achieved at a cut-off threshold value of F(z) ≥ 0.382 in the model (Table 7).

## 4. Discussion

Stable CAD is a chronic condition usually characterized by the accumulation of atherosclerotic plaques in the epicardial arteries. Atherosclerotic plaques in the epicardial arteries can eventually rupture or erode, resulting in acute coronary syndrome in a form of MI or unstable angina. Plaques can also gradually progress to the significant narrowing of the arterial lumen, causing angina pectoris, or they may remain asymptomatic. CAD usually follows a pattern of long stable periods intermitted by unstable episodes due to atherothrombotic events [11].

VWF is essential for platelet adhesion to the subendothelium of the damaged endothelial layer at the high shear rates characteristic of small-diameter arteries, especially at stenotic sites. In vascular wall injury, e.g., endothelial denudation resulting from intravascular intervention, or unstable atherosclerotic plaque rupture, platelets are recruited from the circulation to the subendothelial matrix, forming a hemostatic plug. At high shear rates, platelet adhesion is mediated by the glycoprotein (GP) Ib-IX-V receptor complex located on the platelet surface, as well as by vascular wall collagen and von Willebrand factor [12]. VWF can be considered a bridge between the subendothelium and platelets (Figure 2).

Adhered and activated platelets form sites for the subsequent accumulation of circulating platelets and leukocytes into the injured vessel wall [13]. The release or production of soluble agonists (ADP, thrombin, thromboxane A2, platelet-activating factor, serotonin, etc.) enhances platelet activation and promotes the further accumulation of cells at an injury site. Activated platelets not only participate in thrombus formation, but also initiate and/or accelerate inflammatory processes in the vascular wall [14].

Considering that VWF facilitates primary hemostasis and a local inflammatory response at high shear rates, its dysfunction may contribute to the development of coronary artery disease and its complications. This idea is supported by the observation that CAD is less prevalent in patients with von Willebrand disease (VWD) than in healthy individuals. The authors of [15] assessed the prevalence of atherosclerotic cardiovascular disease (CAD, MI, brain ischemia, and peripheral artery disease) in 7556 patients with VWD and 19,918,970 patients without VWD [15]. They found that the prevalence of CAD was 15.0% in patients with VWD but 26.0% in patients without VWD (*p* < 0.001). They also found that the risk factor-adjusted odds ratio was 0.86 (95% confidence interval (CI), 0.80–0.94) for CAD and 0.69 (95% CI, 0.61–0.79) for MI in patients with VWD [15]. Other studies have found that, in patients with high levels of VWF:Ag, the relative risk of CAD and adverse cardiovascular events was higher than in control groups [16,17,18]. However, in studies where adjustments were made for traditional risk factors, increased WF:Ag levels exhibited no association with CAD and had little effect on the prediction of adverse cardiovascular events [19,20]. In addition, mean levels of VWF:Ag were found by the authors of [21] to be higher in women than in men and increased with age.

In the present study, we focused on exploring VWF:Ag levels, VWF activity (VWF:RCo and VWF:CB), and platelet adhesion to collagen under flow conditions in patients with and without CAD. In patients with and without CAD, VWF:Ag levels and VWF:RCo activity values did not differ. VWF:CB activity was lower in the blood of patients with CAD than in patients without CAD, with values of 106.7% (82.1; 131.6) and 160.4% (112.5; 218.1), respectively (*p* < 0.001). It should be emphasized that the VWF:Ag assay has no differential sensitivities to different molecular weight forms, including large, intermediate, and small multimers, and it detects all multimeric forms of VWF equally well. In contrast, the VWF:CB assay has differential sensitivities to different molecular weight forms, and preferentially detects the large high molecular weight forms of VWF in comparison with the intermediate and small forms [22].

We used a microfluidic device to assess the relative decrease in platelet adhesion after GPIb inhibition. This device simulates blood flow in a small-diameter artery (such as a coronary artery) with a moderate stenosis and exposed collagen. The relative decrease in platelet adhesion after GPIb inhibition was 76.0% (60.6; 82.1) in patients with CAD; this was significantly higher than the figure of 29.3% (0.0; 60.4) recorded for patients in the control group (*p* < 0.001). At the same time, the relative decrease in platelet adhesion after platelet activation inhibition by PGE1 was only 3.9% (−44.0; 51.1) in patients with CAD; this was significantly lower than the figure of 55.8% (40.7; 67.3), recorded for patients in the control group (*p* < 0.001). PGE1 inhibits platelet activation, including the activation of GP IIb/IIIa receptors important for platelet adhesion and aggregation. As platelet activation in patients with CAD was already inhibited by antiplatelet treatment, PGE1 produced little additional inhibitory effect on platelet activation. Of note, most patients without CAD did not receive antiplatelet therapy. The inhibitory effect of PGE1 on platelet adhesion was more pronounced in this group. Contrastingly, the GPIb-mediated component of platelet adhesion was more pronounced in patients with CAD. GPIb are mechanosensing receptors that do not depend on platelet activation [23]. We hypothesize that the GPIb-mediated component of platelet adhesion unaffected by traditional antiplatelet therapy can substantially contribute to platelet adhesion and parietal thrombus formation in patients with CAD.

In the control group, nine patients received only aspirin, and forty-one patients were free from antiplatelet therapy. Aspirin monotherapy had no effect on absolute values of platelet adhesion or values of GPIb-mediated platelet adhesion. All patients with CAD received antiplatelet therapy. The absolute values of platelet adhesion were lower in patients receiving thienopyridines or dual antiplatelet therapy compared with patients receiving only aspirin. Nonetheless, antiplatelet therapy did not affect the GPIb-mediated component of platelet adhesion. As antiplatelet therapy does not affect VWF production or activation or degradation pathways, it cannot affect VWF:Ag, VWF:RCo, or VWF:CB values.

An adjusted odds ratio for CAD was found to be 0.98 (95% CI, 0.97–0.99; *p* = 0.011) per 1% increase in VWF:CB activity, and 1.06 (95% CI, 1.03–1.09; *p* < 0.001) per 1% decrease in platelet adhesion after GPIb inhibition. VWF:CB activity and the relative decrease in platelet adhesion after GPIb inhibition can thus be considered independent risk factors for CAD. Lower VWF:CB activity and more pronounced contribution of VWF to platelet adhesion in patients with CAD, compared with patients without CAD, may indicate a possible critical role played by complex VWF–collagen-platelet interactions in the development of the disease.

This study was not specifically designed to check the etiology hypothesis, and its data cannot directly support or deny it. This study elucidated changes in VWF–collagen-platelet interactions associated with CAD. We suggest that there is a possibility that these changes can be associated with the development of CAD. Nonetheless, further specified research is needed to check the etiology hypothesis.

## 5. Conclusions

In the present study, GPIb-mediated platelet adhesion was more pronounced in patients with than in patients without CAD. We also found that VWF:CB activity was lower in patients with than in patients without CAD. Taken together, these results may indicate a possible critical role played by complex VWF–collagen-platelet interactions in the development of CAD.

## Figures and Tables

**Figure 1 biomedicines-12-02007-f001:**
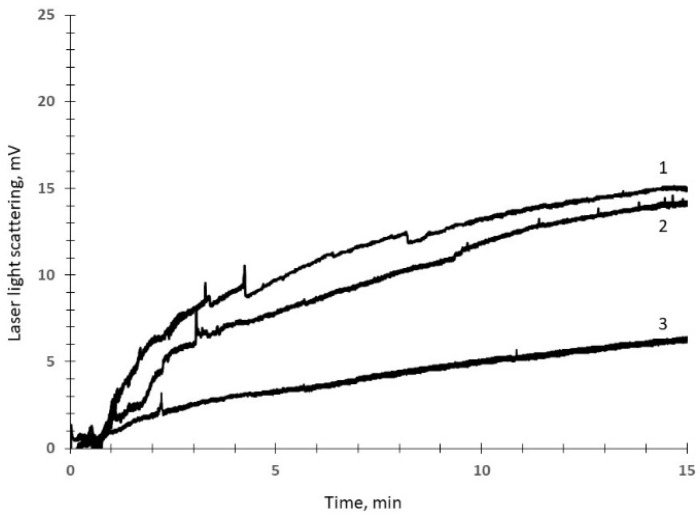
Typical curves of time-dependent increases in the intensity of scattered laser light recorded during 15 min periods of circulation of whole blood samples. Control measurements (1), measurements with 10 µg/mL of glycoprotein Ib (GPIb) inhibition with a monoclonal antibody (2), and measurements with 1 µmol/L of prostaglandin E1 (PGE1) inhibition of platelet activation (3) are shown for a control patient without CAD.

**Figure 2 biomedicines-12-02007-f002:**
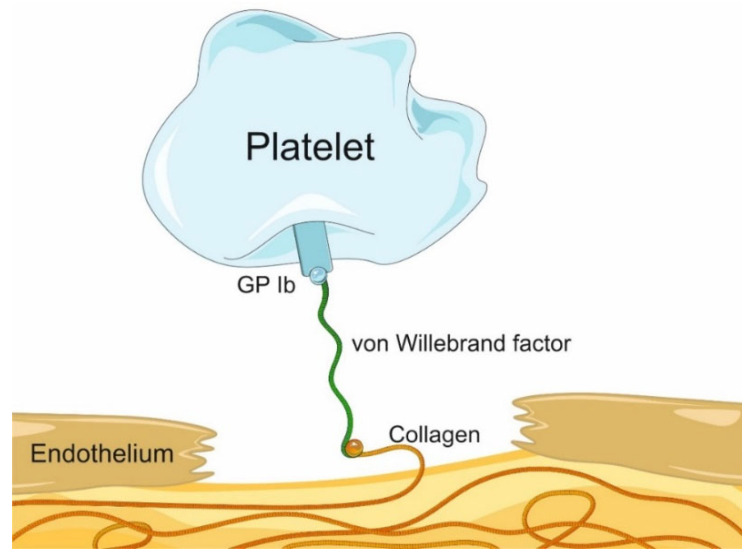
Simplified scheme of von Willebrand factor interaction with platelet glycoprotein Ib-IX-V receptor complex and collagen following endothelium damage.

**Table 1 biomedicines-12-02007-t001:** Clinical characteristics of patients.

Clinical Characteristics	Patients with CAD (n = 30)	Control Group (n = 50)	*p*
Age, years	55 (50; 57)	51 (46; 61)	0.286
Males/females	19 (63%)/11 (37%)	20 (40%)/30 (60%)	0.074
Family history of CAD	9 (30%)	7 (14%)	0.093
Arterial hypertension	25 (83%)	41 (82%)	0.989
Diabetes mellitus	7 (23%)	5 (10%)	0.119
Smoking	21 (70%)	17 (34%)	0.002 *
LDL cholesterol > 3 mmol/L	23 (77%)	30 (60%)	0.314
HDL cholesterol < 1.0 mmol/L for males and <1.2 mmol/L for females	15 (50%)	10 (20%)	0.022 *
Obesity (BMI ≥ 30.0 kg/m^2^)	16 (53%)	21 (42%)	0.452

CAD—coronary artery disease; LDL—low-density lipoproteins; HDL—high-density lipoproteins; BMI—body mass index. * indicates *p* < 0.05 (Mann–Whitney U-test). Data are presented as medians and (first and third) quartiles (Me (Q1; Q3).

**Table 2 biomedicines-12-02007-t002:** Platelet adhesion in patients with CAD and patients in the control group.

Patients	Platelet Adhesion at Baseline, mV	Platelet Adhesion after GPIb Inhibition, mV	Relative Decrease in Platelet Adhesion (∆), %	*p*
Control group	12.6 (9.6; 15.7) *	10.3 (4.3; 14.5) *	29.3 (0.0; 60.4)	<0.001 *
Patients with CAD	7.9 (5.0; 11.7) *	1.8 (1.5; 2.7) *	76.0 (60.6; 82.1)	<0.001 *

CAD—coronary artery disease; GPIb—glycoprotein Ib; ∆—change in platelet adhesion after blocking platelet GPIb receptor mAb compared with its initial value. * indicates comparison of two dependent variables (Wilcoxon test).

**Table 3 biomedicines-12-02007-t003:** Platelet activation inhibition by PGE1 in patients with CAD and patients in the control group.

Patients	Platelet Adhesion at Baseline, mV	Platelet Adhesion after PGE1 Addition, mV	Relative Decrease in Platelet Adhesion (∆), %	*p*
Control group	12.6 (9.6; 15.7) *	5.9 (3.7; 7.9) *	55.8 (40.7; 67.3)	<0.001 *
Patients with CAD	7.9 (5.0; 11.7) *	6.8 (4.7; 10.3) *	3.9 (−44.0; 51.1)	0.50 *

CAD—coronary artery disease; GPIb—glycoprotein Ib; ∆—change in platelet adhesion after platelet activation inhibition by PGE1 compared with its initial value. * indicates comparison of two dependent variables (Wilcoxon test).

**Table 4 biomedicines-12-02007-t004:** VWF antigen levels and activity in patients with CAD and patients in the control group.

Parameters	Patients with CAD (n = 30)	Control Group (n = 50)	*p*
VWF:Ag, % 50–150% normal range	135.2 (108.8; 194.2)	152.0 (114.0; 191.1)	0.58
VWF:RCo, % 50–150% normal range	134.1 (109.0; 185.7)	140.3 (111.8; 175.1)	0.93
VWF:CB, % 50–250% normal range	106.7 (82.1; 131.6)	160.4 (112.5; 218.1)	<0.001

CAD—coronary artery disease; VWF:Ag—von Willebrand factor antigen level; VWF:RCo—von Willebrand factor ristocetin cofactor activity; VWF:CB—von Willebrand factor collagen binding assay; *p*—comparison of two independent variables (Mann–Whitney U-test).

**Table 5 biomedicines-12-02007-t005:** Univariate logistic regression analysis of the relationship between the probability of CAD and selected independent variables.

Variable	Coefficient (β)	OR (95% CI)	*p*
Decrease in platelet adhesion after GPIb inhibition, per 1%	0.049	1.05 (1.03–1.07)	<0.001
VWF:CB, per 1%	−0.016	0.98 (0.97–0.99)	0.002
Smoking	1. 511	4.53 (1.71– 12.02)	0.002
HDL cholesterol < 1.0 mmol/L for males and <1.2 mmol/L for females	1.293	3.64 (1.32–10.04)	0.012
Male sex	0.95	2.59 (1.00–6.69)	0.046
Family history of CAD	0.968	2.63 (0.86–8.05)	0.089
Diabetes mellitus	1.008	2.74 (0.78–9.59)	0.115
Age	0.03	1.03 (0.98–1.09)	0.211
LDL cholesterol > 3 mmol/L	0.622	1.86 (0.66–5.24)	0.239
Obesity	0.456	1.58 (0.64–3.93)	0.326
Arterial hypertension	0.093	1.10 (0.33–3.65)	0.879

CAD—coronary artery disease; GPIb—glycoprotein Ib; LDL—low-density lipoprotein cholesterol; HDL—high-density lipoprotein cholesterol; OR—odds ratio; CI—confidence interval.

**Table 6 biomedicines-12-02007-t006:** Multivariate logistic regression analysis of the relationship between the probability of CAD and independent variables.

Variable	Coefficient (β)	aOR (95% CI)	*p*
Decrease in platelet adhesion after GPIb inhibition, per 1%	0.054	1.06 (1.03–1.09)	<0.001
VWF:CB, per 1%	−0.019	0.98 (0.97–0.99)	0.011
Smoking	1.591	4.91 (1.28–18.79)	0.020
Age	0.084	1.09 (1.01–1.18)	0.037
Intercept	−6.184	0.002	0.019

GPIb—glycoprotein Ib; aOR—adjusted odds ratio; CI—confidence interval.

**Table 7 biomedicines-12-02007-t007:** Receiver operating characteristic (ROC) analysis for the logistic regression model and independent variables.

Variables	AUC	95% CI	*p*
Logistic regression model	91.6% ± 3.1%	85.6–97.6%	<0.001
Decrease in platelet adhesion after GPIb inhibition	82.0% ± 4.7%	72.9–91.1%	<0.001
VWF:CB	27.0% ± 5.6%	16.1–37.9%	0.001
Smoking	68.0% ± 6.2%	55.8–80.2%	0.007
Age	57.1% ± 6.5%	44.5–69.8%	0.288

AUC—area under the curve; CI—confidence interval; GPIb—glycoprotein Ib; VWF:CB—von Willebrand factor collagen binding. Data are presented as means ± SEM.

## Data Availability

The data used in this article are available on request without undue reservation.

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
