# Peer review of "Von Willebrand Factor Collagen-Binding Activity and Von Willebrand Factor-Mediated Platelet Adhesion in Patients with Coronary Artery Disease"

_biomedicines, 2024, doi:10.3390/biomedicines12092007_

Round 1

Reviewer 1 Report

Comments and Suggestions for Authors

In general, although the authors' experiment is necessary for research, the manuscript must undergo some revisions before being accepted for publication.

1. Pay attention to the writing of decimal points, such as line 21 and Table 1. Please check similar problems through the whole manuscript.

2. line 33 & 38: Abbreviations that first appear should be annotated, such as glycoprotein (GP). Please check and revise similar problems throughout the manuscript.

3. What criteria are used to choose sample sizes for patients and control group?

4. The latest incidence rate of coronary artery disease should be introduced in the introduction.

5. The innovation of the article needs to be stated in the last paragraph of the introduction.

6. It is recommended to make some improvements to the title.

Comments on the Quality of English Language

Minor editing of English language required.

Author Response

In general, although the authors' experiment is necessary for research, the manuscript must undergo some revisions before being accepted for publication.

  1. Pay attention to the writing of decimal points, such as line 21 and Table 1. Please check similar problems through the whole manuscript.

Thank you for the precise comment. We have corrected misprints in decimal sign throughout the manuscript (corresponding corrections made in lines 21 & 317, and in Tables 1 & 3).

  1. line 33 & 38: Abbreviations that first appear should be annotated, such as glycoprotein (GP). Please check and revise similar problems throughout the manuscript.

We are sorry for this inaccuracy. We have corrected annotation order of the abbreviations used (corresponding corrections made in lines 33 & 38 for GPIb, and in lines 132 &151 for mAb). We also have double-checked the use of other abbreviations throughout the manuscript. Once again, many thanks for your correction.

  1. What criteria are used to choose sample sizes for patients and control group?

All the patients included into the current study were undergoing regular clinical check-up at the National Medical Research Centre of Cardiology. We aimed to achieve a statistical power of the study 0.8 or higher. By the time when 30 patients with CAD and 50 patients without CAD were enrolled in the study, the statistical power of the study achieved 0.99 for the difference in decrease in platelet adhesion after GPIb inhibition, and 0.92 for the difference in VWF:CB activity between the groups of patients. We considered that the study was powered enough to draw statistically-grounded conclusions and stopped the enrollment.

  1. The latest incidence rate of coronary artery disease should be introduced in the introduction.

            Thank you for a nice suggestion on improvement of the introduction. We have added the following: “Coronary artery disease (CAD) is among the leading causes of disability and mortality worldwide, accounting for 17.8 million deaths annually”

  1. The innovation of the article needs to be stated in the last paragraph of the introduction.

            As you recommended, a statement indicating novelty of our work was added in the end of the “Introduction” section: “To our knowledge, VWF-mediated platelet adhesion and VWF:CB were not investigated previously in CAD patients”

  1. It is recommended to make some improvements to the title.

            We agree that the title in its current form is a bit bulky. On the other hand, we tried to describe the essence of our work as accurately as possible. By now we see only one option how to make the title more concise and not to lose in its specificity – to use abbreviation for VWF. We also removed words ‘to collagen’ after ‘VWF-Mediated platelet adhesion’.

The new title is be as follows: “VWF Collagen-Binding Activity and VWF-mediated Platelet Adhesion in Patients with Coronary Artery Disease”

Comments on the Quality of English Language

Minor editing of English language required.

            As long as none of the authors are native speakers, we have used English language editing service provided by MDPI (confirming certificate will be uploaded along with the revised manuscript). Nevertheless, we assume that some minor errors in language and style could be still present in the text of the manuscript. We are open for any specific suggestions on further language improvements from both the Editors and Reviewers.

Reviewer 2 Report

Comments and Suggestions for Authors

The authors found that coronary artery disease (CAD) patients had lower VWF collagen-binding activity and greater reduction in platelet adhesion after GPIb inhibition compared to controls. These results suggest that VWF–collagen-platelet interactions may be key in CAD development.

General comments

This is a manuscript addressing a topic “Von Willebrand Factor Collagen-Binding Activity and von Willebrand Factor-Mediated Platelet Adhesion to Collagen in Patients with Coronary Artery Disease”. However, the discussion and conclusions drawn are only partly supported by the results. Some concerns need to be addressed.

Specific comments

Major Comments:

1) Lines 324-332: The discussion in this section repeats the main results without sufficient analysis. Specifically, the relative decrease in platelet adhesion after GPIb inhibition was greater, and the relative decrease in platelet adhesion after platelet activation inhibition by PGE1 was smaller in patients with CAD. More in-depth discussion is necessary to address this discrepancy in the results.

2) The change in complex VWF–collagen-platelet interactions may not directly lead to coronary artery narrowing, as less than half of the patients had MI. Additional discussion is needed to explore the correlation between the etiology of the other half of the cases.

3) Since there was a significant difference in antiplatelet medication between the two groups, an explanation of the general effects of these medications on the parameters, including VWF:Ag levels, VWF activity (VWF:RCo and VWF:CB), is required.

Minor Comments:

1) The explanation of the study subjects (those with "chest pain") should also be included in the methods section.

Author Response

Specific comments

Major Comments:

1) Lines 324-332: The discussion in this section repeats the main results without sufficient analysis. Specifically, the relative decrease in platelet adhesion after GPIb inhibition was greater, and the relative decrease in platelet adhesion after platelet activation inhibition by PGE1 was smaller in patients with CAD. More in-depth discussion is necessary to address this discrepancy in the results.

We added the following (please, see lines 343-352): “PGE1 inhibits platelet activation, including activation of GP IIb/IIIa receptors important for platelet adhesion and aggregation. As platelet activation in patients with CAD was already inhibited by antiplatelet treatment, PGE1 produced little additional inhibitory effect on platelet activation. Noteworthy, most patients without CAD did not receive antiplatelet therapy. The inhibitory effect of PGE1 on platelet adhesion was more pronounced in this group. Contrarily, GPIb-mediated component of platelet adhesion was more pronounced in patients with CAD. GPIb are mechanosensing receptors that do not depend on platelet activation [Kim, J. et al. Nature 466, 992–995 (2010). DOI: 10.1038/nature09295]. We hypothesize that GPIb-mediated component of platelet adhesion unaffected by traditional antiplatelet therapy can substantially contribute to platelet adhesion and parietal thrombus formation in patients with CAD.”

2) The change in complex VWF–collagen-platelet interactions may not directly lead to coronary artery narrowing, as less than half of the patients had MI. Additional discussion is needed to explore the correlation between the etiology of the other half of the cases.

Thank you for this remark. Although only half of the patients had MI, all patients with CAD had stenotic lesions, and almost all patients underwent coronary artery stenting (80%) or bypass grafting (13%). Nonetheless, this study was not specifically designed to check the etiology hypothesis, and its data cannot directly support or deny it. This study elucidated changes in VWF-collagen-platelet interactions associated with CAD. In the paper, we suggest that there is a possibility that these changes can be associated with the development of CAD. Nonetheless, further specified research is needed to check the etiology hypothesis.

We added this commentary in the lines 386-390.

3) Since there was a significant difference in antiplatelet medication between the two groups, an explanation of the general effects of these medications on the parameters, including VWF:Ag levels, VWF activity (VWF:RCo and VWF:CB), is required.

The following was added (lines 370-372): “As antiplatelet therapy does not affect VWF production, activation or degradation pathways, it cannot affect VWF:Ag, VWF:RCo or VWF:CB values”

Minor Comments:

1) The explanation of the study subjects (those with "chest pain") should also be included in the methods section.

The following was added to description of patients with CAD: “These patients had a history of or typical symptoms of CAD” (line 74) and patients without CAD: “who underwent CAG due to other diagnostic reasons” (line 77). Definition of ‘patients with chest pain’ was removed from the text to prevent ambiguity.

Round 2

Reviewer 2 Report

Comments and Suggestions for Authors

The authors have corrected the manuscript according to the reviewer's comments.